# The macroeconomics of abortion: A scoping review and analysis of the costs and outcomes

Yana van der Meulen Rodgers[1,2]*, Ernestina Coast[3], Samantha R. Lattof[3], Cheri Poss[4], Brittany Moore[4]

1 Department of Labor Studies and Employment Relations, Rutgers University, Piscataway, New Jersey, United States of America, 2 Department of Women's and Gender Studies, Rutgers University, Piscataway, New Jersey, United States of America, 3 Department of International Development, London School of Economics and Political Science, London, United Kingdom, 4 Ipas, Chapel Hill, North Carolina, United States of America

* yana.rodgers@rutgers.edu

**Data Availability Statement:** All relevant data are within the manuscript and its Supporting Information files.

**Funding:** This work was supported by the Netherlands Ministry of Foreign Affairs in the form

## Abstract

### Background

Although abortion is a common gynecological procedure around the globe, we lack synthesis of the known macroeconomic costs and outcomes of abortion care and abortion policies. This scoping review synthesizes the literature on the impact of abortion-related care and abortion policies on economic outcomes at the macroeconomic level (that is, for societies and nation states).

### Methods and findings

Searches were conducted in eight electronic databases. We conducted the searches and application of inclusion/exclusion criteria using the PRISMA extension for Scoping Reviews. For inclusion, studies must have examined one of the following macroeconomic outcomes: costs, impacts, benefits, and/or value of abortion care or abortion policies. Quantitative and qualitative data were extracted for descriptive statistics and thematic analysis. Of the 189 data extractions with macroeconomic evidence, costs at the national level are the most frequently reported economic outcome (n = 97), followed by impacts (n = 66), and benefits/value (n = 26). Findings show that post-abortion care services can constitute a substantial portion of national expenditures on health. Public sector coverage of abortion costs is sparse, and individuals bear most of the costs. Evidence also indicates that liberalizing abortion laws can have positive spillover effects for women's educational attainment and labor supply, and that access to abortion services contributes to improvements in children's human capital. However, the political economy around abortion legislation remains complicated and controversial.

### Conclusions

Given the highly charged political nature of abortion around the global and the preponderance of rhetoric that can cloud reality in policy dialogues, it is imperative that social science

of funds awarded to YvdMR, EC, SRL, CP, and BM (activity number 28438). The funder had no role in study design, data collection and analysis, decision to publish, or preparation of the manuscript.

**Competing interests:** The authors have declared that no competing interests exist.

researchers build the evidence base on the macroeconomic outcomes of abortion services and regulations.

## Introduction

In many countries, restrictive abortion laws, the inability of individuals to afford abortions in the private sector, and a shortage of healthcare workers willing or able to perform abortions have all contributed to insufficient access to safe abortion services [1]. Even when abortion is legal, social norms and attitudes are often slower to change than legal statutes, so people may still obtain clandestine abortions. Additionally, there are often regulatory or infrastructure barriers that can restrict access to care. In the face of these barriers, the widespread dissemination of information through the internet has helped to destigmatize both abortion and contraception, and it has provided healthcare practitioners and abortion seekers with clinical information about birth control and safe abortion procedures, including medical abortion [2]. In some countries, changes in abortion access and practices have contributed to increasing abortion rates despite highly restrictive national legislation. Hence the legal status of abortion does not always reflect access to care due to the effects of regulations, stigma, and medical abortion.

The financial costs to society of abortion restrictions, the extent to which the public sector covers the cost of abortion services, and the spillover effects of abortion access on women's educational attainment and employment rates are examples of the macroeconomic costs and outcomes of abortion services and abortion laws. This paper synthesizes a large body of social science evidence on the macroeconomics of abortion and articulates a set of key themes around abortion costs, impacts, and benefits of abortion services (including un/safe abortion care and post-abortion care) and abortion regulations. We examine the evidence base and also identify evidence gaps on the costs and benefits of abortion to stakeholders at the macroeconomic level, which encompasses entire societies and nation states. Results from our microeconomic and mesoeconomic analyses and a discussion of the role of stigma are presented in separate companion articles.

To achieve this objective, the scoping review answers the following question: What are the macroeconomic costs, impacts, and/or benefits of abortion care and abortion policies? This paper explores how access to abortion services and changes in abortion laws affect broad aggregates such as women's labor supply and educational attainment, indicators of societal well-being such as crime, investment in children's human capital, and national income. Knowledge of such themes provides a better understanding of the overcall context in which individuals seek abortions and the extent to which they are affected by, and in turn have an influence on, macroeconomic aggregates and the multiple channels through which those outcomes occur.

## Methodology

We took a systematic approach to finding evidence on the economics of abortion by conducting a scoping review that includes as widely as possible all of the relevant literature. Like systematic reviews, scoping reviews use a systematic approach to searching, screening, and reporting [3]. Following the Preferred Reporting Items for Systematic reviews and Meta-Analyses (PRISMA) scoping review extension (PRISMA-ScR) and reporting guidelines [4], we developed a protocol (Coast *et al.* [5]) to ensure our review was manageable, transparent, and reproducible.

**Table 1. PICOTS criteria used in the scoping review.**

| PICOTS | |
|---|---|
| Populations | Societies and nation states in which individuals obtain abortions or post-abortion care |
| Interventions | Induced abortion (safe/unsafe), post-abortion care, and/or abortion policies |
| Control | None |
| Outcomes | Quantitative or qualitative data on: <br> • macroeconomic costs of abortion care or abortion policies <br> • macroeconomic impacts of abortion care or abortion policies <br> • macroeconomic benefits of abortion care or abortion policies <br> • macroeconomic value of abortion care or abortion policies |
| Timeframe | 1 September 1994 to 15 January 2019 |

The scoping review considered articles on induced abortion and/or post-abortion care in any world region that were published in peer-reviewed journals (see PICOTS in Table 1). These data must have covered one of the following four outcomes of abortion care or abortion policies: cost (the amount paid by national governments to cover abortion services or financial results of abortion policies; impact (the macro-level effect or influence of abortion care or abortion policies); benefit (advantages at the national level from receiving abortion care or implementing abortion policies); and value (the importance, worth, welfare gains, or utility of receiving abortion care or implementing abortion policies).

We excluded policy briefs, books, book chapters, editorials, commentaries, and published or unpublished reports from governments and other agencies. By limiting the sample of studies to peer-reviewed articles that have been subjected to the scrutiny of other experts in the field, we minimized the likelihood of including studies with errors [6]. We did not make any assessments of the quality of included items since the objective of this scoping review is to synthesize and describe the coverage of the evidence.

We chose eight electronic databases for searching: Cumulative Index to Nursing and Allied Health (CINAHL); EconLit; Excerpta Medica Database (EMBASE); International Bibliography of the Social Sciences (IBSS); JSTOR; PubMed; ScienceDirect; and Web of Science. These sources were searched using combinations of relevant search terms (Table 2) that we developed and tested for sensitivity in advance of the scoping review.

Quantitative and qualitative data were extracted for descriptive statistics and thematic analysis using an inductive approach. Our analysis synthesizes the evidence base and identifies evidence

**Table 2. Search terms and their combinations.**

| 1. Abortion terms | 2. Economic terms | 3. Impact terms |
|---|---|---|
| abort* | cost* | cost* |
| termination of pregnancy | econom* | benefit* |
| terminate pregnancy | price* | value* |
| pregnancy termination | financ* | impact* |
| pregnancy terminations | resource* | |
| Postabortion | fee* | |
| post-abortion | tax* | |
| | expenditure* | |
| | GDP | |
| | gross domestic product | |
| | pay* | |
| | expens* | |

gaps on the macroeconomic costs and impacts of abortion care. Data are reported using a systematic narrative synthesis in which the results are presented narratively and organized thematically, supplemented with tables of descriptive statistics on included studies and their outcomes.

## Findings

### Descriptive statistics

As shown in Fig 1, the search generated 19,653 items for screening. After duplicate removal, the 16,918 remaining items were screened for inclusion on the basis of title and abstract (TIAB). We determined eligibility of all items, and unclear items were discussed. Where exclusion could not be determined on the basis of TIAB, the authors screened the full text. Decisions were made in favor of an inclusive approach where questions remained. In total, 782 articles went through the process of a full text screening, where we screened on article type, intervention, outcomes, and timeframe. After this final step, 158 studies with macroeconomic content met all the inclusion criteria and were included in the scoping review.

Among the countries covered in the 158 studies, a large number examined the United States. Almost one half of all the studies (71 out of 158) focused exclusively on the United States, and an additional 5 studies on a selected group of countries included the United States in their analysis (Table 3). This dominance of the United States in studies of abortion not only reflects the political attention on abortion, but also the availability of data, the institutional affiliation of authors, the editorial home of journals, and the location of funding and other resources for conducting studies.

After the United States, the country with the most coverage was the United Kingdom (n = 7), followed by India (n = 5). Interestingly, more studies have focused on African countries relative to countries in Asia, even though a larger share of the world's population resides in Asia. Relatively few studies have focused on countries in Latin America and the Caribbean, most likely because abortion policies are more restrictive in this region than any other part of the world. These restrictions would also serve as obstacles facing researchers. Noticeably absent with just one exception (Israel) are countries in the Middle East and North Africa.

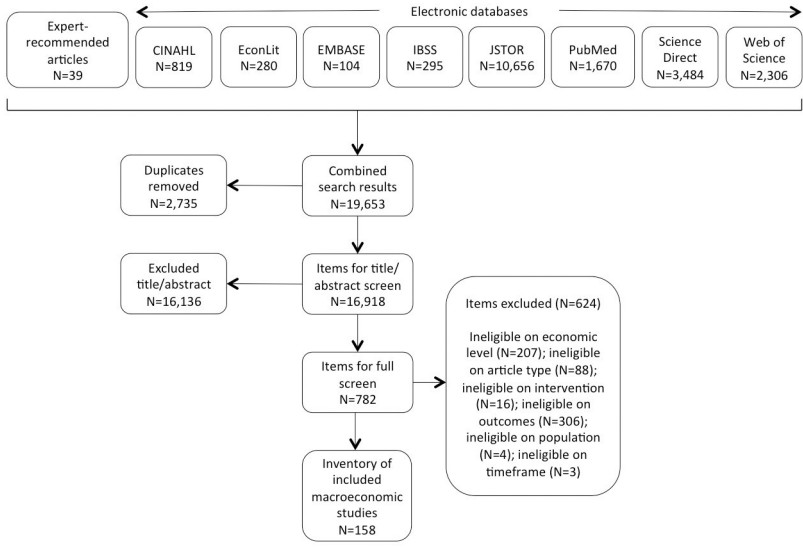

**Fig 1. Screening results.**

**Table 3. Included studies by region and country.**

| Region/country | # of studies | Region/country | # of studies |
|---|---|---|---|
| **Northern America** | **74** | **Europe** | **15** |
| Canada | 3 | Ireland | 2 |
| United States | 71 | Moldova | 1 |
| | | Romania | 2 |
| **Africa** | **21** | United Kingdom | 7 |
| Ethiopia | 1 | Multiple countries | 3 |
| Ghana | 2 | | |
| Malawi | 1 | **Latin America & Caribbean** | **10** |
| Nigeria | 2 | Argentina | 1 |
| Rwanda | 1 | Colombia | 2 |
| South Africa | 5 | Mexico | 3 |
| Uganda | 3 | Puerto Rico | 1 |
| Zambia | 1 | Multiple countries | 3 |
| Multiple Countries | 5 | | |
| | | **Oceana** | **2** |
| **Asia** | **15** | Australia | 2 |
| Bangladesh | 2 | | |
| China | 1 | **Cross-Regional Studies** | **21** |
| India | 5 | Global | 14 |
| Israel | 1 | Selected countries incl. US | 5 |
| Kazakhstan | 1 | Selected countries excl. US | 2 |
| Nepal | 1 | | |
| Taiwan | 1 | | |
| Thailand | 1 | | |
| Vietnam | 1 | | |
| Multiple countries | 1 | **Total** | **158** |

The majority of studies were quantitative in nature, with 106 studies relying exclusively on quantitative methods and another 32 studies including both quantitative and qualitative methods (Table 4). Many (47) of the studies used regression analysis. Two-thirds of the lead authors were women, which may not come as a surprise given that the topic is reproductive health and the study populations are often women of reproductive age. Studies ranged in their level of geographic coverage, with over half of the studies examining abortion outcomes at the national level. Those studies that were conducted at the sub-national level and below still passed the screening criteria for inclusion because they included information at the macro level.

In total, we had 189 extractions of data at the macro level, which exceeds the number of screened articles (158) because some articles provided information on multiple indicators. In examining this data, we explored the financial costs of abortion services, the extent to which these costs are borne by national health systems, how abortion law regulations affect women's human capital and that of their children, and the political economy of abortion restrictions and liberalization. Table 5 reports the themes that emerged from the synthesis of extracted data. Note that to facilitate the analysis, we merged studies on benefits and value. Each of the themes is discussed in detail below.

**Table 4. Characteristics of included studies (n = 158).**

|  | *No. Studies* |
|---|---|
| **Type of Data** |  |
| Quantitative | 106 |
| Qualitative | 20 |
| Both | 32 |
| **Methodology** |  |
| Cohort analytic | 3 |
| Cohort (before & after) | 3 |
| Interrupted time series | 1 |
| Qualitative | 9 |
| Mixed methods | 14 |
| Regression | 47 |
| Literature review | 21 |
| Other | 60 |
| **Presumed Gender of 1st Author** |  |
| Woman | 103 |
| Man | 46 |
| Unclear | 9 |
| **Geographical Level** |  |
| National | 86 |
| Sub-national (e.g. state, city) | 26 |
| Local (e.g. village) | 3 |
| Health facility | 14 |
| Other | 29 |
| **Study Population** |  |
| Ethnic (or race) | 2 |
| National | 38 |
| Religion | 1 |
| Geographical location (e.g. urban/rural, region, facility) | 20 |
| Socio-economic | 1 |
| Age (e.g. adolescents) | 10 |
| Status as abortion seeker | 15 |
| Multiple answers from list | 30 |
| Other, specify | 33 |
| Abortion provider | 8 |

## Macroeconomic costs

Our scoping review resulted in 97 articles containing data on the macroeconomic costs of abortion care services and policies (Table 5; S1 Appendix). The process of synthesizing this knowledge generated the following four major themes.

**(1) Post-abortion care services are expensive and absorb a large portion of government health budgets.** Unsafe abortion is a major public health problem, especially in contexts where access to legal abortion is highly restricted. An estimated 7.9 percent of maternal deaths are due to unsafe abortion [7]; unsafe abortion is also a leading cause of maternal morbidity. Globally, almost 300,000 maternal deaths occur each year [7, 8]. In a comprehensive analysis of maternal mortality around the world using data from 2003 to 2009, Say *et al.* [7] found that unsafe abortion accounts for a higher percentage of maternal mortality in Latin America and

**Table 5. Macroeconomic content and themes of included studies (n = 189).**

| Themes | n |
|---|---|
| **Studies on economic costs** | **97** |
| 1. Post-abortion care services constitute a large portion of government health budgets | 17 |
| 2. Public-sector coverage of abortion care services is sparse | 39 |
| 3. Restrictive abortion laws impose financial hardships and often result in costly delays | 24 |
| 4. Some countries are actively working to provide more cost-effective and innovative options | 17 |
| **Studies on economic impact** | **66** |
| 1. Abortion laws have spillover effects on women's educational attainment and labor supply | 19 |
| 2. Access to abortion services affects human capital investment of the next generation | 8 |
| 3. Abortion law liberalization may lead to lower crime rates | 7 |
| 4. Political economy around abortion law is complicated | 32 |
| **Studies on economic benefits or value** | **26** |
| 1. Selection effects of abortion law liberalization are associated with other long-term benefits for children | 4 |
| 2. Sex-selective abortions are a large industry in some countries | 2 |
| 3. Integrating abortion into full set of reproductive health services has societal benefits | 10 |
| 4. Abortion care services can have macro-level benefits that are not well categorized | 10 |

the Caribbean than in any other region, followed closely by Sub-Saharan Africa. About 10 percent of all maternal mortality in Latin America and the Caribbean is caused by unsafe abortion. This region also has the most restrictive national abortion legislation in the world [9]. In contrast, unsafe abortion is found to cause less than one percent of maternal mortality in East Asia, a region with higher income and less restrictive abortion laws.

In countries with laws and policies restricting abortion access, post-abortion care is often prioritized as a way to address the issue of unsafe abortion. At least 50 developing countries had formal programs for post-abortion care as of 2018 [10]. The total cost of post-abortion care to public health systems in many countries is substantial. In particular, Vlassoff *et al*. [11] estimate that US$171 million is spent annually to treat abortion complications in Africa. Additional estimates for a sample of developing countries indicate that the total cost of post-abortion care per case ranges from $334 in Rwanda to $972 in Colombia, which constitutes up to 35 percent of annual per capita income depending on the country [12]. Governments would save money by refocusing their efforts on providing greater access to modern contraception and to safe abortion options. For example, South Africa's public health sector could save over $28 million in a 10-year period if women had greater access to safe abortion services [13].

On average, the cost of meeting a woman's unmet need for modern contraceptive supplies and services for one year would amount to just 3 to 12 percent of the average cost of treating a patient who requires post-abortion care [12]. In Ethiopia, the annual cost of providing post-abortion care at the national level totals $47 million per year, which comprises a substantial portion of total national expenditures on reproductive health [14]. Using a similar methodology, Vlassoff and his coauthors estimate this cost to be $20.8 million per year in Uganda [15] and $2.5 million per year in Rwanda [16].

Studies by other authors also find high post-abortion care costs. In Zambia, post-abortion care following an unsafe abortion can cost the health system 2.5 times more than safe abortion care [17]. Other countries for which relatively high post-abortion care costs have been documented include Malawi [18], Nigeria [19], South Africa [20], Colombia [21, 22], and Bangladesh [23, 24]. Furthermore, two studies for Latin American countries have found that treating post-abortion complications comprises over half of the countries' public sector budgets for obstetric care [25, 26].

**(2) Public sector coverage of abortion care services is sparse, and individuals often bear most of the financial costs.** Very few countries have public sectors that fully cover the financial costs of obtaining an abortion, resulting in large numbers of individuals globally who are forced to pay out-of-pocket for abortions. Making sure that people have access to financial resources to cover these costs is crucial in countries that have banned abortion so that individuals can access medically-supervised abortion services [27]. In countries where abortion is illegal, public health services may absorb the cost of post-abortion care, often in relatively expensive tertiary-level hospitals, but they do not cover access to safe abortions [28]. Closely related, a number of studies have estimated cost savings if more safe abortions were provided by the public sector. A related strain of this literature estimates cost savings to society from abortion services that have reduced the need for public spending on the medical and welfare costs of pregnancies carried to term among low-income women. For example, Murthy and Creinin [29] estimate that in the United States, every dollar that is spent on abortions for low-income women translates into $4 in savings that would have been spent on prenatal care, delivery services, and medical assistance over the next two years.

Those countries that do use public sector funding to cover all or a large portion of the costs of abortion tend to be high-income economies, including Australia [30], Canada [31] and the United Kingdom [32, 33]. Overall, of all countries with liberal abortion laws, 34 countries have full public funding for abortion, 25 countries have partial funding, and 21 countries have no funding at all or only under exceptional cases [34]. A disproportionate share of countries that provide full public funding are high-income countries. Among developing countries, Nepal stands out for having recently instituted a set of safe abortion guidelines that includes free abortion services at all government facilities [35]. India (where abortion is legal) also provides free abortion services in the public sector, although some states have allowed public providers to start charging various user fees or to have private practices [36]. In some countries (such as South Africa) there may be state funding for abortion, but shortages of trained health personnel and insufficient technology mean that state-funded abortions are not available at most primary health facilities and local health clinics [37].

Several studies on the United States have examined how state-level restrictions on Medicaid funding–the need-based system of public health insurance–affect access to abortion and the affordability of abortion services. These state-level restrictions come on top of the Hyde Amendment at the national level, which restricts the use of federal funds to pay for abortions. Abortion is the only reproductive health service that is not covered by federal Medicaid [38]. Some states place further restrictions on the use of private insurance to pay for abortion services. Studies comparing states across the country generally find that state-level restrictions on Medicaid are correlated with lower abortion rates in states that have such restrictions [39–41], and they contribute to individuals' resource constraints in seeking abortion services [42, 43]. Research also suggests that abortion-related restrictions on public funding for family planning services are costly on a broader scale: without publicly-funded family planning services, states would be spending more than $1.2 billion annually in their Medicaid programs to cover the costs of unplanned births [44].

**(3) Restrictive abortion laws impose financial hardships on abortion seekers, often resulting in costly delays.** Since 2010 the United States has seen an unprecedented expansion in state-level policies that restrict abortion access. These policies can be grouped into six major types of restrictive abortion laws enacted by states, each of which the Supreme Court has found to be constitutional: (1) restrictions on Medicaid funds to pay for abortions, (2) parental involvement laws that require unmarried teen minors to obtain parental consent or require providers to notify the minor's parent before an abortion is performed, (3) mandatory counseling laws that require providers to give patients state-mandated medical information

about possible risks and side effects of abortion at least 24 hours before the procedure, (4) two-visit laws that require that abortion seekers receive the mandatory counseling materials in person, (5) Targeted Regulation of Abortion Provider laws (TRAP laws) that impose on abortion providers a variety of burdensome staffing and physical plant requirements not imposed on other clinics performing comparable medical services, and (6) laws banning a late-term abortion with dilation and extraction.

The most common of these state-level restrictions are mandatory counseling laws. Survey data indicate that over half of women in the Midwest and about two-thirds of women in the Northeast did not disagree with the idea of having a mandatory counseling law [45]. That said, survey respondents said they were more negatively impacted by state-level abortion restrictions than positively impacted, with the most common negative impacts including delays in receiving care, longer travel, and time away from work [45].

Abortion restrictions have aimed to reduce the overall demand for abortions by those seeking services as well as the overall supply of providers. A major channel through which they do so is to reduce accessibility and effectively raise the price of abortion, which is associated with fewer reported abortions [46, 47]. Anti-abortion activities by protesters such as picketing and blocking patients can have similar effects, with one set of estimates indicating that anti-abortion activities in the United States have raised the price of abortion by about 4 percent and reduced the overall abortion rate by 19 percent [48]. There is some variation across studies in the demand-side effects of abortion restrictions. For example, estimates of the aggregate demand for abortion services by minors suggest that in the United States, parental involvement laws reduce the aggregate demand for abortion services by minors anywhere from 13 to 25 percent [49]. Similarly, Medoff [50] finds that both parental notification laws and mandatory counseling laws increase the price of abortion and result in a decline in the aggregate demand for abortion by 7 to 14 percent. Lack of public funding can cause women to delay seeking abortion care until their pregnancies have advanced into the second trimester as they seek out financial resources to access abortion services [51, 52].

On the supply side, evidence in Calkin [53] indicates that state-level abortion restrictions have contributed to clinic closures and reduced operations. At the time of the study, there were 27 American cities considered to be "abortion deserts" because women had to travel over 100 miles to reach the nearest abortion clinic. The inability to travel such distances to reach abortion providers disproportionately affects low-income women who are both income and time constrained. Evidence indicates that 89 percent of American counties have no abortion providers within their county borders [54].

**(4) Some countries are actively working to provide more cost-effective and innovative options for abortion seekers.** The past few decades have seen improvements in reproductive-health technologies, the availability of online information on reproductive health, and the ability to order abortion pills online. A growing body of evidence indicates that the increased availability and affordability of misoprostol has made medical abortion more common. Not only do individuals often choose the cheaper option, but so do health insurance companies and public sector health officials. Because medical abortion does not usually require physicians with surgical skills or surgical facilities, medical abortion is often considered the cheaper option at the national level and more health systems are likely to move to medical abortion in the future [55]. Exceptions to this argument include Nigeria, where clinic-based manual vacuum aspiration is more cost effective than medical abortion [56].

Additional studies have looked at the macro-level cost savings of alternative abortion technologies, including manual vacuum aspiration and medical abortions. For example, in Mexico, greater access to manual vacuum aspiration and medical abortion could save the government up to US$ 1.6 million annually, with further cost savings if more safe abortion services were

provided in outpatient settings at smaller public and private health facilities [57]. Similarly large savings to the government from providing greater access to manual vacuum aspiration and medical abortions are found for Colombia [22].

Some countries have experimented with medical abortion via telemedicine and hotlines to reach abortion seekers who may otherwise be harder to reach. This option was tried in Australia [58] and in several Latin American countries [59]. Some countries including the UK are also moving toward regulatory reforms in which nurses and midwives are allowed to perform more abortion services in order to provide cost savings to national health systems [60].

## Macroeconomic impacts

Our scoping review resulted in 66 articles containing data on the macroeconomic impacts of abortion care services and policies (Table 5; S2 Appendix). Most of these studies are on the broader economic impacts of imposing abortion regulations or liberalizing abortion laws. This knowledge can be synthesized into four general themes.

**(1) Abortion regulations have spillover effects on women's educational attainment and labor supply.** Abortion regulations can act to raise the price of abortion and reduce aggregate demand for abortion [50, 61–63]. Women's fertility rates and even marital rates are also impacted by abortion regulations. In Eastern Europe, regulations that made abortion more accessible led to a large decrease in births [64] and an increase in marriage rates for non-teenage women [65]. In the United States, birth rates increased in states with more restrictive policies and higher contraception costs [66]. Further evidence indicates that states that funded abortion services had lower birth rates among teenagers [67]. A study of long-term trends in the United States confirms that the legalization of abortion has contributed to a decline in childbearing (largely due to an increase in the number of childless women) and to women's economic progress [68]. The effects of abortion access were even stronger than access to the birth control pill in driving women's decisions to delay marriage and childbearing [69].

A number of studies have linked the legalization of abortion to women's advancement in education and in the labor market. In particular, Kalist [70] found that by reducing unwanted births, legalization of abortion in the United States led to increased labor force participation rates for women, especially for single black women. A similar result was found in Angrist and Evans [71], with substantial increases in high school graduation, college attendance, and employment for black women who were teenagers when state abortion laws were liberalized. Additional evidence indicates that women who were denied an abortion because of restrictive state laws not only were less likely to be employed full time, they were also more likely to live in poverty and to require public assistance compared to women who obtained abortions [72]. Bloom *et al.* [73] took this point about women's employment one step further and found that lower fertility (instrumented by the legalization of abortion) increases women's labor supply and contributes positively to GDP growth.

**(2) Access to abortion services affects the human capital investment of the next generation.** The legalization of abortion is also linked to various measures of investment in children's human capital. In the United States, children born after the Supreme Court's 1973 Roe v Wade ruling were more likely to graduate from college and less likely to be welfare recipients or single parents [46]. Children's outcomes may have improved on average because they were more likely to be born into a household in which they were wanted. Follow-up research indicates that these kinds of effects on cohort characteristics for children were different depending on whether the births were avoided due to abortion access versus access to the birth control pill. In particular, Ananat and Hungerman [74] find that the effect of abortion access on the living situation of an average child is smaller than the effect of access to the pill, largely because

pill access increased the likelihood of a child having a college-educated, married, "upwardly-mobile" mother. Other evidence indicates that abortion legalization led to a sharp decrease in unintended births, which in turn is associated with increased schooling and greater earnings of children born after abortion legalization [75]. Some of these effects may have a race dimension, with results in Whitaker [76] pointing to a stronger association between abortion rates and high school graduation rates for young black men than for other demographic groups.

There is also evidence on abortion and children's human capital from outside of the United States. In particular, Romania's abortion ban is associated with worse educational outcomes and labor market achievements of children born after the ban [77, 78]. And in Sub-Saharan Africa, abortion law liberalization is linked to greater parental investment in girls' schooling, with the rationale that access to abortion lowers the likelihood of a girl child dropping out of school in the event of an unplanned pregnancy [79]. In Taiwan, legalization of abortion as well as sex-detecting technology also impacted girls' education. Girls born at a higher birth order were born into families where they were wanted and where parents invested in their educations. The outcome was an increase in university attendance for girls born after the legalization of abortion [80].

**(3) Abortion law liberalization may lead to lower crime rates.**  Perhaps most famously among studies on abortion in the economics literature, the legalization of abortion has been linked to crime reduction. In a widely-cited study for the United States, Donohue and Levitt [81] found that crime rates across states appear to have dropped as a result of Roe v. Wade. The main channel through which this outcome occurred is that children who were born unwanted in low-income households in underprivileged communities before the legalization of abortion grew up to be at greater risk of engaging in crime as adults. Abortion liberalization reduced the incidence of such unwanted births, with a subsequent reduction in crime years later when those babies would have been adults. This study has prompted several article-length critiques [82, 83] disputing the findings based on the construction of age cohorts, data on crime rates, and assumptions about abortion rates before Roe v. Wade. These critiques in turn were countered with adjusted estimates from Donohue and Levitt showing that abortion legalization does have a causal effect in reducing crime [84, 85]. However, the finding may not be generalizable to other countries. Buonanno *et al*. [86] show that in Europe, abortion liberalization is not associated with reduced crime, most likely because strong welfare systems and family ties serve to minimize the risk that unwanted childbearing results in greater crime [86].

**(4) The political economy around abortion law is complicated and controversial.**  From the complete criminalization of all abortions to abortion upon request, legislation on abortion varies substantially across countries. Countries with no restrictions are largely found in the Global North, while countries with the most restrictions tend to be in the Global South, especially in Latin America and the Caribbean and in Sub-Saharan Africa. As of 2019, abortion was completely prohibited or only permitted to save a woman's life in 65 countries, accounting for 27 percent of the world's population [87]. In contrast, 66 countries accounting for about 36 percent of the world's population allowed individuals to have an abortion without restriction as to the reason [87]. Legislation not only varies geographically, it has slowly changed over time [88]. Between 1994 and 2019, almost 50 countries liberalized their abortion laws, with some countries overturning their complete abortion bans and other countries incrementally allowing more reasons (such as health and economic necessity) for which individuals may obtain an abortion [87].

It is not feasible to cleanly summarize the political economy of these legislative changes, given that most studies uncovered by our scoping review focus on individual countries, each with particular contexts and political environments. Several studies have pointed to the high costs of post-abortion care as a key impetus to motivate legislative reforms that improve access

to safe abortions [11, 17, 25, 28]. Other studies have pointed to weaknesses in national health systems in providing cost-effective abortions and the need to reform health system financing and to strengthen public abortion services [13, 23, 27, 36, 89]. Overall the evidence makes clear that the countries that have seen the most rapid change in women's health indicators are those that have changed their laws while simultaneously improving their service delivery.

## Macroeconomics benefits and value

Our scoping review resulted in 26 articles containing data on the macroeconomic benefits and value of abortion care services and policies (Table 5; S3 Appendix). Four studies included data specific to the selection effect of abortion liberalization (through which abortion can help to prevent unwanted births) and additional benefits to children born after liberalization, especially their increased chances of living in better economic circumstances [46, 72, 90, 91]. For example, Foster *et al.* [90] compared women who received abortions with women who were denied abortions due to state regulations and found that women who were able to delay childbirth until they had greater economic and emotional security were able to have closer relationships with their children and raise them in relatively better economic circumstances, with fewer indicators of delayed child development [90].

Two articles had macro-level data on sex-selective abortions and their value as an industry as well as the influence of institutional factors. In India, the process of using ultrasound technology to determine the sex of a fetus had become a 5 billion rupee industry by 2005, reflecting growing demand for sex determination as well as a burgeoning sex-selective abortion business [92]. Institutional factors such as the ability of couples to rely on the national pension system for old-age support as well as India's strong education gradient play an important role in sex selection and son preference [31].

Some countries have moved more actively toward integrating safe abortion services into a full continuum of reproductive health services. Even when they are legally allowed, abortion services are often separated from other health services. This separation has only served to marginalize individuals who seek abortions and has made it more difficult to reduce the incidence of unsafe abortions. Adopting a broader strategy aimed at providing an integrated package of services for reproductive health care has much potential to reduce global inequities in maternal mortality. Ten articles included data on this theme.

Evidence for India [93] indicates that such an approach is more effective in terms of lives saved and is more cost effective when compared to a non-integrated strategy without family planning and safe abortion services. In Mexico, including access to safe abortion into a policy strategy designed to reduce maternal mortality would save about $116 million over the lifetime of a birth cohort [94]. Even more important than the cost savings to national governments is lives saved. In Sub-Saharan Africa, transitioning from unsafe to safe abortion is estimated to result in 18,300 (Nigeria) and 29,000 (Ghana) years of life gained per 100,000 safe abortion procedures. And in the United States, societal benefits from increasing access to abortion services through state Medicaid coverage for medically-necessary abortion services is associated with fewer cases of severe maternal morbidity compared to states without such coverage [95].

The remaining articles with extracted data on benefits and values had findings that did not overlap in content or key themes with other articles or with each other. These studies range from the need for abortion services among migrant workers, to the biopolitical interest of the state in having a healthy population [96, 97]. This final set of articles underscores the point that abortion care services can have macro-level benefits and values that may not be well known, easily observed, or readily categorized, but they still merit attention.

## Conclusion

This article has presented the results of a scoping review of the social science literature on the macroeconomic costs and outcomes of abortion-related care and abortion policies. Our review shows that there is a wealth of economically-relevant information that can be gleaned from the evidence base. At the macro level, post-abortion care services are expensive and can constitute a substantial portion of government health budgets. Public sector coverage of abortion costs is sparse, and abortion seekers bear most of the financial costs. In a diverse range of contexts, restrictive abortion regulations impose financial hardships on individuals, often resulting in costly delays. The interplays between legal restrictions on access to abortion and delays to abortion-related care are indeed striking. By unpacking the points at which abortion laws contribute to delays to abortion-related care, we can achieve greater insight into the health and economic costs for individuals and for entire societies that result from these delays. By further exploring the intersectionality of these economic factors, we can better understand the ways in which health systems and societal structures reproduce injustices and inequities. This scoping review has also highlighted evidence that the impacts of abortion regulations on fertility can have strong spillover effects on women's educational attainment and labor supply. Moreover, access to abortion services appears to contribute to improvements in children's human capital after the liberalization of abortion laws, often due to a selection effect in which abortion can help to prevent unwanted births.

Our review indicates that from a disciplinary and methodological perspective, there are methodological gaps. The evidence base has an abundance of studies using regression analysis, while behavioral economics approaches are underexploited. Moreover, there are gaps in country coverage, with a disproportionate amount of the evidence base focused on the United States. Given the highly charged political nature of abortion around the global and the preponderance of rhetoric that can cloud reality in policy dialogues, it is imperative that social science researchers consider using a more diverse set of methodological tools and cover a wider range of countries as they build the evidence base on the macroeconomic outcomes of abortion services and regulations.

## Supporting information

**S1 Checklist.**
(DOCX)

**S1 Appendix. Summary of studies reporting macroeconomics costs (n = 97).**
(DOCX)

**S2 Appendix. Summary of studies reporting macroeconomics impacts (n = 66).**
(DOCX)

**S3 Appendix. Summary of studies reporting macroeconomics value/benefit (n = 26).**
(DOCX)

## Acknowledgments

We wish to thank Elaine Zundl (EZ), Lisbeth Gall (LG), and Joe Strong (JS) for their assistance with screening and data extraction.

## Author Contributions

**Conceptualization:** Yana van der Meulen Rodgers, Ernestina Coast, Cheri Poss, Brittany Moore.

**Data curation:** Samantha R. Lattof.

**Formal analysis:** Yana van der Meulen Rodgers, Ernestina Coast, Samantha R. Lattof, Cheri Poss, Brittany Moore.

**Funding acquisition:** Yana van der Meulen Rodgers, Ernestina Coast, Cheri Poss, Brittany Moore.

**Investigation:** Yana van der Meulen Rodgers, Ernestina Coast, Samantha R. Lattof, Cheri Poss, Brittany Moore.

**Methodology:** Ernestina Coast, Samantha R. Lattof.

**Project administration:** Cheri Poss, Brittany Moore.

**Visualization:** Yana van der Meulen Rodgers, Samantha R. Lattof.

**Writing – original draft:** Yana van der Meulen Rodgers.

**Writing – review & editing:** Ernestina Coast, Samantha R. Lattof, Cheri Poss, Brittany Moore.

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
