## [Decision Letter · Decision Letter 0]

13 Apr 2021

The macroeconomics of abortion: A scoping review and analysis of the costs and outcomes

PONE-D-20-01142

Dear Dr. Rodgers,

We’re pleased to inform you that your manuscript has been judged scientifically suitable for publication and will be formally accepted for publication once it meets all outstanding technical requirements.

Kind regards,

M. Mahmud Khan

Academic Editor

PLOS ONE

Journal Requirements:

1. Please complete a PRISMA-ScR checklist (available at https://www.equator-network.org/wp-content/uploads/2018/09/PRISMA-ScR-Fillable-Checklist-1.docx) and upload it as supplementary file.

2. Thank you for stating the following financial disclosure:  [no statement provided].

Please provide an amended Funding Statement that declares *all* the funding or sources of support received during this specific study (whether external or internal to your organization) as detailed online in our guide for authors at http://journals.plos.org/plosone/s/submit-now

Please state what role the funders took in the study.  If any authors received a salary from any of your funders, please state which authors and which funder. If the funders had no role, please state: "The funders had no role in study design, data collection and analysis, decision to publish, or preparation of the manuscript."

c. Please send your amended statements by return email; we will change the online submission form on your behalf.

Reviewers' comments:

Reviewer's Responses to Questions

**Comments to the Author**

1. Is the manuscript technically sound, and do the data support the conclusions?

Reviewer #1: Yes

2. Has the statistical analysis been performed appropriately and rigorously? 

Reviewer #1: Yes

3. Have the authors made all data underlying the findings in their manuscript fully available?

Reviewer #1: Yes

4. Is the manuscript presented in an intelligible fashion and written in standard English?

Reviewer #1: Yes

5. Review Comments to the Author

Reviewer #1: The topic is interesting and it represents a long term actual subject with multiple medical and social implications. The manuscript is a very well written and the research is detailed.

The conclusions are formulated in concordance with the results.

The manuscript seems ready for publication.

6. PLOS authors have the option to publish the peer review history of their article (what does this mean?). If published, this will include your full peer review and any attached files.

Reviewer #1: No

---

## [Editor Report · Acceptance letter]

23 Apr 2021

PONE-D-20-01142 

The macroeconomics of abortion: A scoping review and analysis of the costs and outcomes 

Dear Dr. Rodgers:

I'm pleased to inform you that your manuscript has been deemed suitable for publication in PLOS ONE. Congratulations! Your manuscript is now with our production department. 

Kind regards, 

on behalf of

Dr. M. Mahmud Khan 

Academic Editor

PLOS ONE